

# The potential for geostationary remote sensing of NO₂ to improve weather prediction

Xueling Liu[1], Arthur P. Mizzi[2,*], Jeffrey L. Anderson[3], Inez Fung[1] and Ronald C. Cohen[1,4]

[1]Department of Earth and Planetary Science, University of California at Berkeley, Berkeley, CA, USA

[2]Atmospheric Chemistry Observation and Modeling Laboratory, National Center for Atmospheric Research, Boulder, CO, USA

[3]Institute for Mathematics Applied to Geosciences, National Center for Atmospheric Research, Boulder, CO, USA

[4]Department of Chemistry, University of California at Berkeley, Berkeley, CA, USA

[*] Now an NCAR/DAREs Visitor and at NASA Ames Research Center

*Correspondence to*: Ronald C. Cohen (rccohen@berkeley.edu)





**Abstract.** Observations of winds in the planetary boundary layer remain sparse making it challenging to simulate and predict atmospheric conditions that are most important for describing and predicting urban air quality. Short-lived chemicals are observed as plumes whose location is affected by boundary layer winds and with a lifetime affected by boundary layer height and mixing. Here we investigate the application of data assimilation of $NO_2$ columns as will be observed from geostationary orbit to improve predictions and retrospective analysis of wind fields in the boundary layer.

## 1 Introduction

Data assimilation methods are a fundamental tool for numerical weather prediction (NWP) with observations of temperature, pressure, winds, humidity, etc. used as constraints on initial conditions and time evolution of atmospheric energy and winds (e.g. Bauer et al., 2015). With the exception of water vapor and ozone (e.g. Inness et al., 2019), observations of atmospheric constituents are generally not used in current NWP. However, the tools of data assimilation are increasingly the focus of a variety of off-line chemical transport models (CTM) that aim to improve regional air quality forecasts and to enhance understanding of emissions of gases and aerosol into the atmosphere (e.g. Miyazaki et al. 2014, 2017, 2020 Bocquet et al., 2015; Zhang et al., 2012; Lahoz et al., 2007) and there is growing interest in on-line assimilation of other chemicals (e.g. Dee et al. 2014, Gelaro et al. 2012). Meteorology and chemical constituents are not independent. Coupled chemistry meteorology models such as WRF-Chem have evolved rapidly in recent years (Grell and Baklanov, 2011). This development in numerical modeling offers the opportunity to study the interaction/feedback between atmospheric physics, dynamics and composition such as the impact of air constituents on incoming radiation, the modification of weather (cloud formation and precipitation) by natural and anthropogenic aerosol, and the impact of climate change on the frequency and strength of events with poor air quality (e.g. Fiore et al., 2012; Grell et al., 2011). In parallel with this advance in modeling capability, observations of gases and aerosols from space based instruments are providing an unprecedented view of constituents from the surface to the mesosphere. Space observations of column $NO_2$ have been applied in the verification of point-source emissions (e.g. Russell et al. 2012; Beirle et al., 2011), quantification of uncertain sources (include biogenic and soil emissions) (Lin, 2012), and detection of episodic events, such as wildfire and lighting ( Zhu et al. 2019; Miyazaki et al., 2014; Mebust et al., 2011).

The combination of these two advances sets the stage for joint assimilation of both meteorology and chemistry where chemical observations can improve the representation of dynamical motions in the atmosphere. In concept, it is easy to see the potential benefit of assimilating composition observations. For example, modeled winds might be transporting material to the southwest while an observed plume is moving to the west. In this example, the chemical observations would cause the assimilated model to alter the wind direction and thus, aligning the predicted meteorology with the observed flow of chemicals. This is just one example. Chemical observations are also sensitive to wind speed (Laughner, et al. 2016; Valin, et al. 2013;) and PBL dynamics. Examples of the beneficial information flow across the two sub-systems include the improvement in cloud distributions after assimilating aerosols (Saide et al., 2012) and the potential for improvement in temperature, winds and cyclone development during dust storms via



assimilation of aerosol optical depth (AOD) (Reale et al., 2011, 2014). Improvement in stratospheric winds by assimilating chemical tracers has also been demonstrated (Allen et al., 2013; Chu et al., 2013; Milewski and Bourqui, 2011; Semane et al., 2009; Peuch et al., 2000). Examples in simpler models include studies by Allen et al., 2014, 2015; and Haussaire and Bocquet, 2015. Among the challenges that must be addressed as we begin to understand the potential benefits of joint assimilation of physical state variables and composition are the aspects of two linked sub-systems (meteorology and chemistry) that can be most efficiently improved by linking them to observed chemical fields.

Aerosols, CO and $CO_2$ have been the focus of most prior chemical assimilation ( Mizzi et al., 2016; Saide et al., 2014; Liu et al., 2012;). In our first analysis of $NO_2$ assimilation (Liu et al., 2017) we examined the potential for assimilation of high spatial (~ 3 km) and temporal (hourly) resolution $NO_2$ columns as will be provided by future geostationary observations to improve the representation of $NO_x$ emissions. $NO_x$ has a lifetime of order five hours within the boundary layer and thus exhibits variation of concentrations at the spatial scales of order 50~75 km downwind of emissions. Those fine temporal and spatial scales make $NO_x$ variations more strongly coupled to short-timescale meteorological parameters than other more long-lived chemical tracers such as aerosol or CO. In our initial research (Liu et al., 2017), we focused on the retrieval of the $NO_x$ emissions. We found that using the column $NO_2$ to constrain emission accurately required simultaneous meteorology and chemical assimilation. The strongest constraints were found in regions with high emissions and using hourly assimilation of meteorological observations.

Our assimilation anticipates the launch of a Geostationary satellite for column $NO_2$ observations, Tropospheric Emissions: Monitoring of Pollution (TEMPO) scheduled for launch in 2022. Related instruments include The Korean GEMS instrument launched in early 2020 and the ESA Sentinel 4 instrument to be launched in in the near future. The TEMPO observations will have two features that will make them a significant advance compared to current instruments in low earth orbit. First, the instrument will make measurements with hourly repeats during the sunlit portion of the day. Second the instrument will have approximately 3×3 km pixels, a substantial increase in spatial resolution compared to the OMI instrument and an improvement over the TROPOMI instrument (Zoogman et al. 2017). That spatial resolution is also sufficient to quantify gradients in $NO_2$ that result from the combined effects of emissions, chemistry and transport.

Here we focus on winds. We expect the influence of $NO_2$ column assimilation on wind fields to be at the spatial scale of 75 km set by the $NO_2$ chemical lifetime (e.g. Laughner and Cohen 2019) and the average wind speed. We begin by describing the data assimilation tools and a simulator for future geostationary satellite observations of $NO_2$ columns (section 2). In section 3 we describe assimilation experiments that provide insight into the constraints that the column $NO_2$ observations will have on winds. In section 4, we discuss the improvements to the accuracy of the modelled winds and assess the potential benefits of this approach to data assimilation. We conclude in section 5.

## 2 Methodology

The data assimilation system is comprised of the forecast model WRF-Chem and the Data Assimilation Research Testbed (DART) as described in Mizzi et al., (2016) and Liu et al., (2017). The WRF-Chem/DART setup, TEMPO simulator and meteorological observations are described in more detail in





Liu et al., (2017). Here we briefly describe the updated data assimilation system that allows $NO_2$ observations to influence winds.

## 2.1 WRF-Chem model

We use WRF-Chem version 3.7 with a one-way nested domain (Figure 1). The outer domain of 12 km resolution covers the western United States and the inner domain of 3 km resolution covers Denver and the mountain region on its west. On the outer domain the initial and boundary conditions are driven by weather reanalysis data for meteorology and by MOZART for chemistry (Emmons et al. 2010). After one-month simulation on the outer domain, the inner domain is initialized with the initial and boundary conditions taken from the outer domain simulations.

The anthropogenic emissions are taken from the National Emission Inventory (NEI) 2011, which describes the hourly emissions for a typical summertime weekday. Biogenic emissions are parameterized using Model of Emissions of Gases and Aerosols from Nature (MEGAN) (Guenther et al. 2006). Gas phase reactions are simulated using regional acid deposition model version 2 (RADM2).

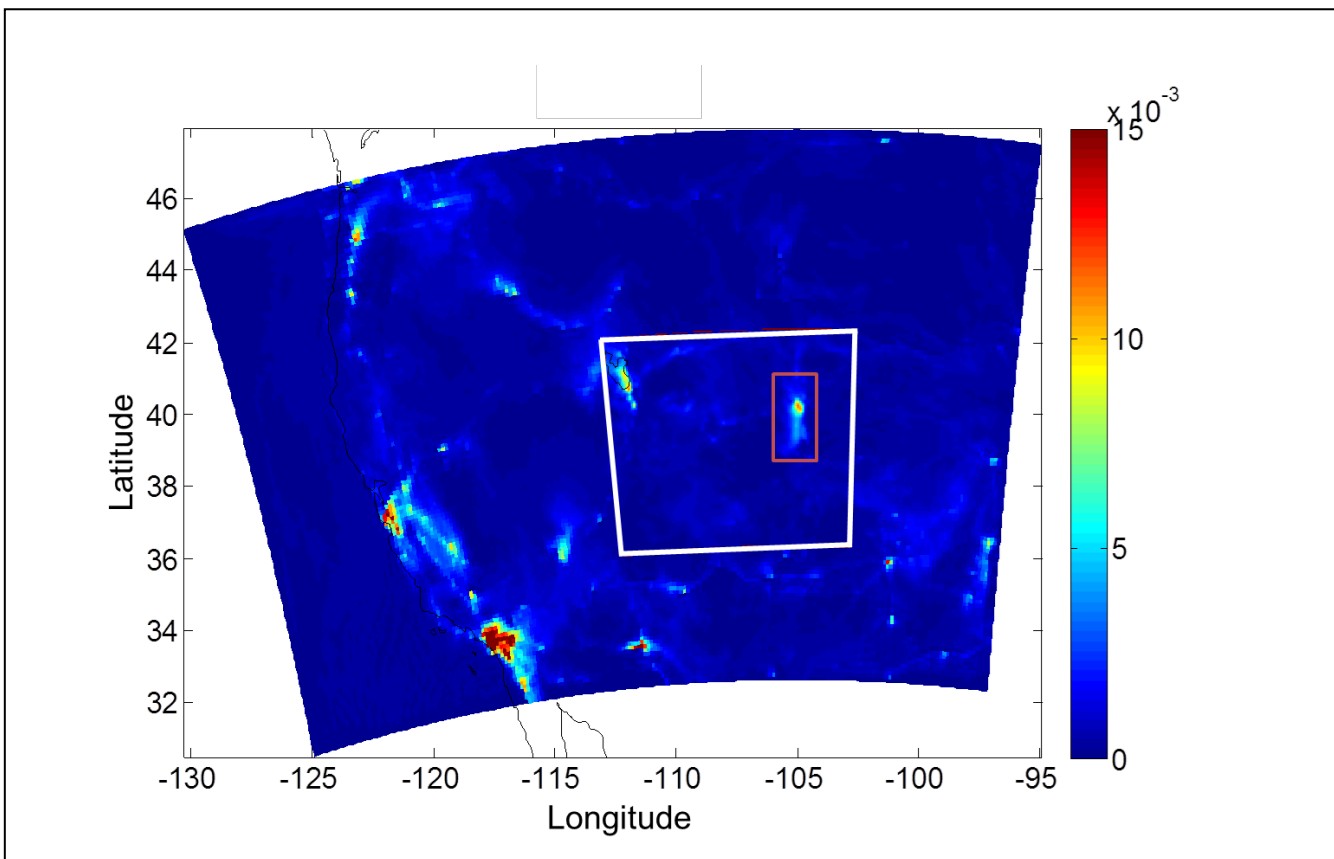

**Figure 1.** Model domain is 12 km outer domain and 3 km inner domain. Data assimilation is performed on the inner domain.

## 2.2 DART assimilation system

WRF-Chem-DART is a regional multivariate data assimilation system developed by the National Center
for Atmospheric Research (NCAR) to analyse meteorological and chemical states simultaneously
(Anderson and Collins 2007, Anderson et al. 2009). In this study we use the DART toolkit configured as
the ensemble adjustment Kalman filter (EAKF) (Anderson 2001). We apply adaptive spatially and
temporally varying inflation to the prior state to maintain the ensemble spread (Anderson et al. 2009). We
use horizontal localization to reduce influence from spurious correlations (Anderson 2012). A Gaspari
and Cohn weighting function is applied with weights diminishing to zero 20 km away from the
observation location. As in Liu et al. (2017), sensitivity tests show that $NO_2$ data assimilation with an





hour assimilation window performs the best using the weighting function with a width of 20 km. The analysed chemical states are $NO_2$ concentrations. The analysed meteorological states include U, V, W, T, QVAPOR, QCLOUD, QRAIN, QICE, QSNOW, MU, PH, T2, Q2, U10, V10 and PSFC as described in Romine 2013 Table 2. The analysis is updated using DART from continuously cycled one-hour 30 member ensemble WRF-Chem forecasts. The DART configuration details are provided in Liu et al 2017.

Previous studies that assimilate chemistry and meteorology simultaneously apply the variable localization approach (Arellano 2007, Kang 2011, Liu 2017) which zeroes out the covariance between chemistry and some of the meteorology variables without taking advantage of the information related to meteorology carried by the chemical tracers. In this study, we partially turn off the variable localization and allow the assimilated $NO_2$ observations to influence horizontal wind (U and V). With this setup, the advection scheme in the WRF-Chem model predicts downwind $NO_2$ evolution based on the wind fields. The EAKF computes the covariances between the predicted $NO_2$ distribution and wind variables. These sensitivities are utilized to refine the model state toward one that best fits the $NO_2$ observations considering the confidence in both the observations and model prediction.

### 2.3 Initial and boundary condition ensembles

We add random perturbations to the temperature field of a single initial state to produce an ensemble of perturbed meteorological initial conditions. The perturbations were generated by sampling the NCEP background error covariance using the WRF Data Assimilation System (WRFDA) (Barker 2012) with cv_option=3. The statistics are estimated with the differences of 24 and 48-hour GFS forecasts with T170 resolution, valid at the same time for 357 cases, distributed over a period of one year. The ensemble member lateral boundary conditions perturbations are generated using the DART pert_wrf_bc program based on the initial ensemble. The DART update_wrf_bc program is used to update the boundary condition including the tendency for the analysis time to match the analysis states from DART.

### 2.4 Observations

To generate synthetic TEMPO $NO_2$ retrievals, we use the TEMPO $NO_2$ simulator developed in Liu et al. (2017) as the observation operator to compute the observed column from a model prediction. It includes a layer dependent Box-Air Mass Factor (BAMF) for each observation pixel. BAMF is atmospheric scattering weights that depend on parameters including viewing geometry, surface (terrain or cloud) pressure, and surface reflectivity. The parameters used to compute BAMFs are sampled from a model run with hourly frequency and clear sky conditions. Details for the TEMPO simulator and observation error generation are described in (Liu et al., 2017).

### 3 Assimilation experiments

We perform observing system simulation experiments (OSSE) to analyze the wind constraints from synthetic $NO_2$ observations. We initialized the WRF-Chem nature run (NR) on a 12 km resolution domain (d01) at 2014060100. The meteorological initial and boundary conditions are taken from the North



American Mesoscale Forecast System (NAM), and the chemistry simulation is constrained by MOZART output. After one-month simulation on d01, the NR on the 3 km domain (d02) is initialized from the d01 model simulation at 2014070215. Its meteorological and chemical boundary conditions are provided by NAM reanalysis data and the d01 simulation respectively. We have a parallel model simulation labelled control run (CR), which is performed in the same way as the NR except its meteorological simulation is initialized and constrained by NARR. Constrained by different reanalysis data, the NR- and CR-simulated winds in the boundary layer differ and thus show the discrepancy in the $NO_2$ transport processes. We perform data assimilation on d02 from 2014070313 to 2014070600 with an hourly assimilation window. In our OSSE, the nature run (NR) simulations are considered as the "true atmosphere" from which synthetic $NO_2$ observations are generated using the TEMPO simulator. After a one-hour forecast the prior ensemble is combined with synthetic $NO_2$ observations to produce the posterior ensemble. The difference in wind and $NO_2$ simulations between the NR and the ensemble mean results from the utilization of two different reanalysis data as meteorological constraints and from the assimilation while we apply the same forecast model, emission input and model physics/chemistry scheme. The posterior ensemble will be used as the initial conditions to forecast the next hour. We evaluate the data assimilation performance by comparing the mean of the posterior estimate with the NR simulations.

We designed a series of six experiments to evaluate the potential of geostationary observations of column $NO_2$ to improve wind fields. First, we conduct a free model run (FREE) with 30 ensemble members derived from the CR without data assimilation. This will set the baseline performance and will be compared with cases that assimilate observations to evaluate the benefit of data assimilation to improve the winds. In the second experiment (CHEM), we assimilate TEMPO $NO_2$ observations over the 12-hour daytime to constrain the winds in the ensemble. By comparing with FREE, we can evaluate the improvement in wind simulations as a result of assimilating $NO_2$ observations. In experiment (T, H), we assimilate hourly observations of temperature and humidity which can indirectly update winds via the covariances of temperature and humidity against wind states. In experiment (T, H, CHEM) we assimilate TEMPO $NO_2$ observations together with temperature and humidity observations. In this case, wind analyses are constrained by the multiple indirect observations via covariances with temperature, humidity and $NO_2$. In the experiment (MET), we assimilate all meteorological observations including wind, temperature and humidity. This is representative of the current weather observing systems representation of boundary layer winds. Finally, we assimilate TEMPO $NO_2$ observations in addition to the meteorological observations in the experiment (MET, CHEM) to assess the influence of $NO_2$ observations on winds under the circumstances of a full meteorology assimilation.

**4 Results and discussion**

We compare the assimilation results with the NR states to evaluate the assimilation performance. The RMSE of the observed quantities are calculated as $\sqrt{\sum_i^n (y_i^m - y_i^t)^2 / n}$, where $y_i^m$ and $y_i^t$ are the model and true values for the $i$th observation, respectively, and $n$ is the total number of observations located within a sub-model space in Figure 2. The RMSE of the model states are calculated as $\sqrt{\sum_i^l (x_i^m - x_i^t)^2 / l}$, where $x_i^m$ and $x_i^t$ are the model and true values at the $i$th model grid point, respectively, and $l$ is the total





number of grid points of interest. For the analysed wind variables, the grid points of interest are all the points located within a model sub-domain as shown in Fig. 2, containing the lowest 5 model levels vertically. We find that the horizontal transport of $NO_2$ is most sensitive to the winds in the lowest 5 model levels, and the top of the shallow boundary layer in the morning is as low as the 5th model level.

We also analyse the uncertainty (spread) of the prior and posterior estimates. The uncertainty is calculated as the 1-σ standard deviation of the ensemble.

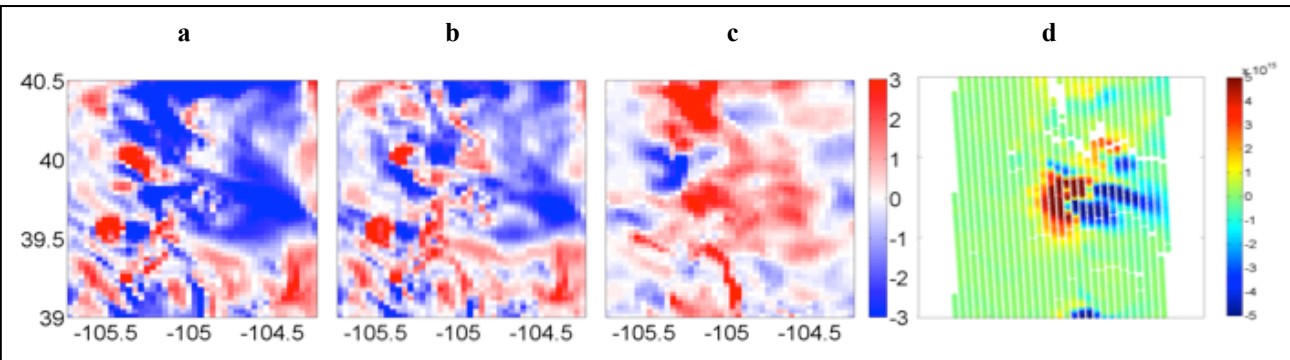

**Figure 2:** The U wind state variable at 13:00 MST on July 4th **a.** prior minus truth **b.** posterior – truth **c.** posterior minus prior and **d.** the difference between the prior TEMPO $NO_2$ column and the truth.

## 4.1 $NO_2$ assimilation

The performance of ensemble-based assimilation is determined by the representation of the ensemble uncertainty. In OSSEs we test how well the ensemble system represents the uncertainty by comparing the ensemble spread with the RMSE computed with respect to the true observations. Figure 3 shows the temporal evolution of the RMSE and the spread for TEMPO $NO_2$ column observations in FREE and the three experiments with TEMPO observations assimilated. We find that in all experiments the variation of

prior ensemble spread follows the fluctuations of the prior RMSE with similar magnitude after the first day of assimilation. This indicates that the ensemble system develops a good amount of spread for $NO_2$ states and wind states as well, because the $NO_2$ spread results from the wind differences among ensemble members.

For all the experiments assimilating TEMPO $NO_2$ observations, the diurnal variation of the prior RMSE

and spread is related to the $NO_2$ column variation with the peaks in the morning and evening rush hours and local minima in the early afternoon. The errors in the comparison to the TEMPO $NO_2$ columns are reduced by 78 % on average from the prior to the posterior estimates. The temporal average of the posterior RMSE varies from 2.6 to $2.9\times10^{14}$ molecules/cm$^3$, which is very similar to the $NO_2$ assimilation results in our previous experiment ENS.1 (Liu, et al. 2017, Figure 4). Experiment CHEM shows lower

prior RMSE of TEMPO $NO_2$ than the FREE for two reasons. First, assimilation of TEMPO in CHEM





reduces the errors in the posterior $NO_2$ of the last cycle, which results in better forecast prior $NO_2$. Second, assimilation of $NO_2$ improves the wind forecast in models (as shown in Section 4.2) and thus reduces the $NO_2$ transport errors. This demonstrates that in places without wind observations, assimilating TEMPO $NO_2$ observations can reduce the errors in the $NO_2$ forecast by allowing $NO_2$ observations to improve wind simulations in models.

**Figure 3a.** Evaluation of the time evolution of TEMPO column $NO_2$ observations in Denver from 2 July 10:00 to 5 July 18:00 for the CHEM experiment Prior (black) and posterior (red). **Top** RMSE **Bottom** Spread.



**Figure 3b.** Evaluation of the Time evolution of pseudo TEMPO $NO_2$ observations in Denver from 2 July 10:00 to 5 July 18:00 for T,H + CHEM experiment. Prior (black) and posterior (red). **Top** RMSE **Bottom** Spread.

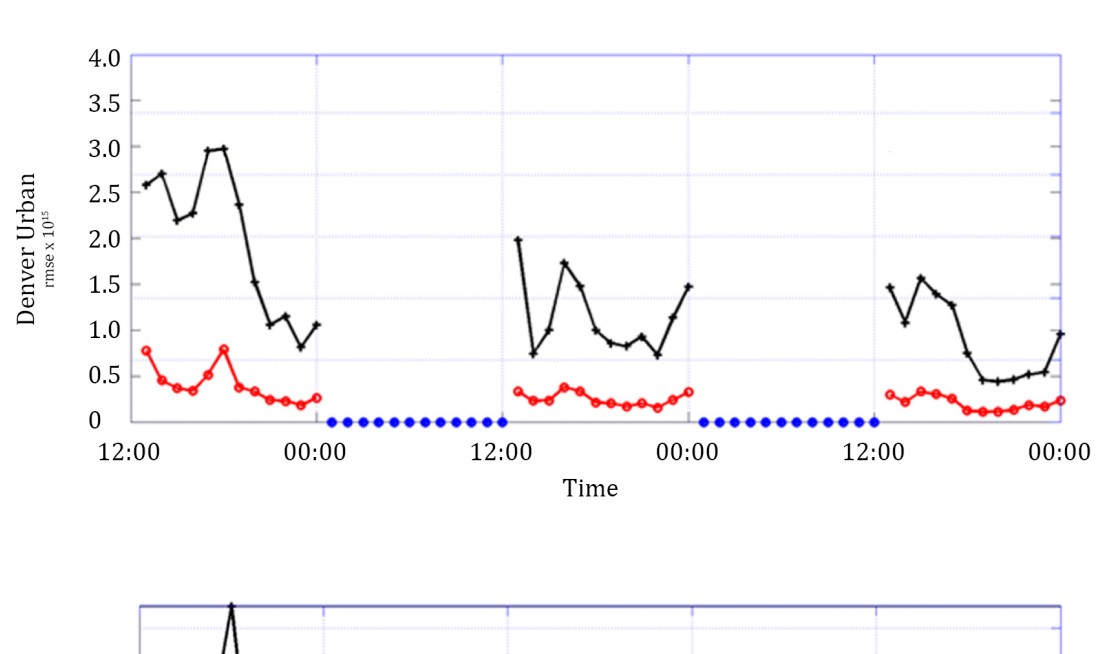

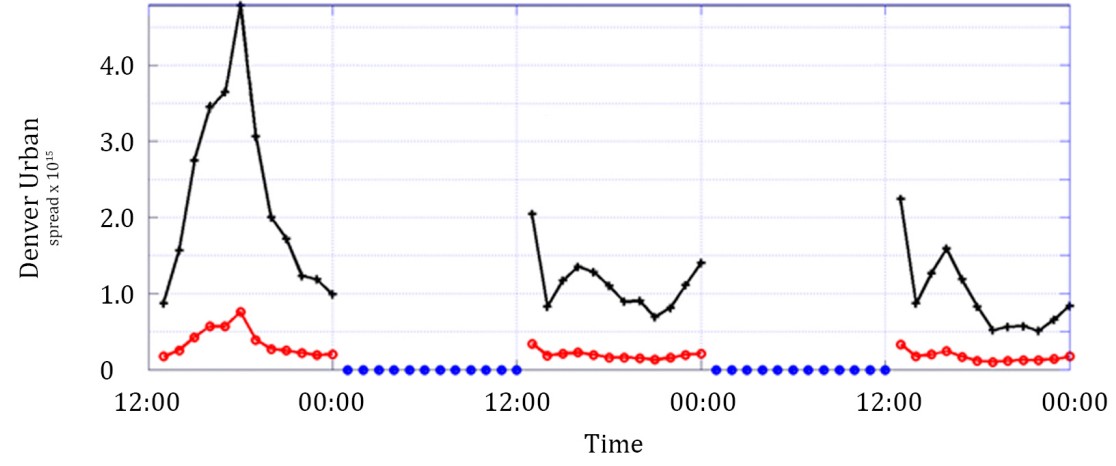





**Figure 3c.** Evaluation of the Time evolution of pseudo TEMPO NO$_2$ observations in Denver from 2 July 10:00 to 5 July 18:00 for MET+CHEM experiment. Prior (black) and posterior (red). **Top** RMSE **Bottom** Spread.

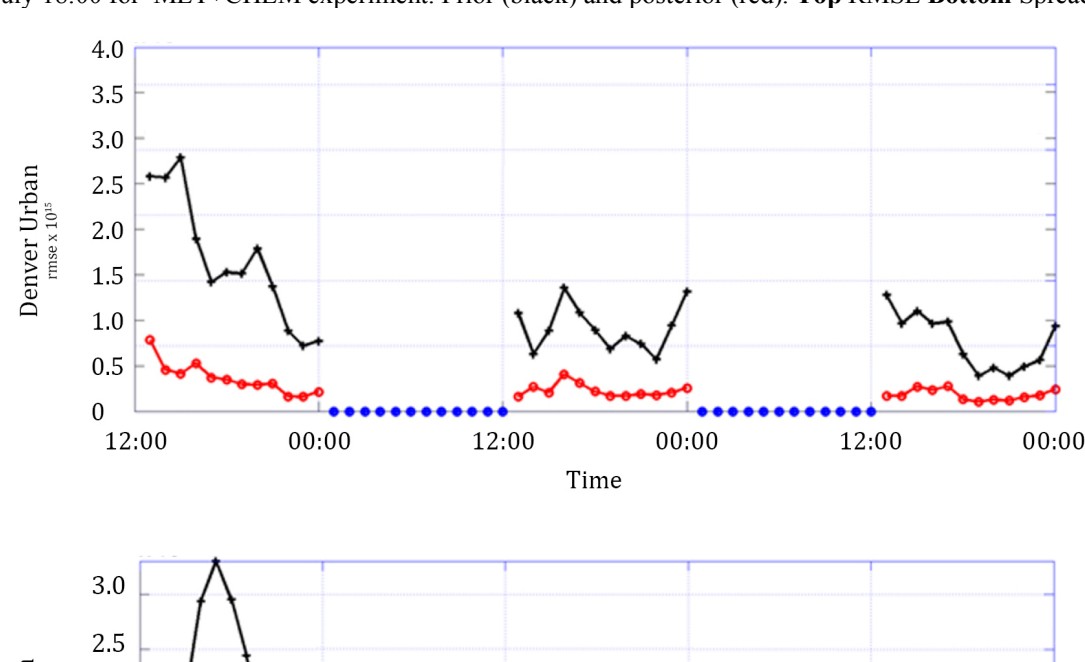

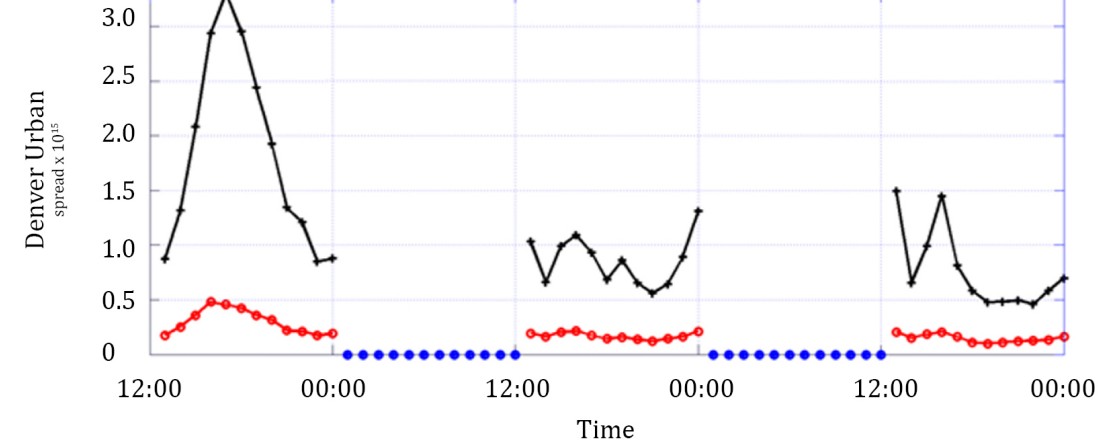



## 4.2 Using TEMPO NO$_2$ observations to constrain the winds

Errors of the winds in models affect horizontal advection of NO$_2$ and result in differences between observed and modeled NO$_2$ vertical column density that can be used to correct the winds. In this ensemble assimilation system, we examine the impact of assimilating TEMPO NO$_2$ observations on the winds in
the boundary layer when different sets of meteorological observations are assimilated. Figure 4 shows the hourly evolution of the posterior RMSE of wind state variable U for all 6 experiments. Results for V are similar. We exclude the first daytime point in our analysis because it takes time for the assimilation system to equilibrate. Without any constraint on winds, FREE shows varying wind RMSE with higher values in the night than the daytime. With the assimilation of TEMPO only, CHEM shows error reduction in the
posterior wind analysis in each daytime cycle. Figure 5 compares the temporal average of the posterior wind RMSE for the 6 runs during daytime. The daytime average posterior RMSE is reduced by 0.44 m/s (15.70 %) and 0.41 m/s (15.45 %) for U and V wind from FREE to CHEM. We find that the reduction in wind RMSE resulting from daytime assimilation disappears after the first night cycle (Figure 4). This is because the daytime error reduction is only observed in regions with abundant NO$_2$ concentrations; wind
error in regions with little NO$_2$ remain high during the day, and quickly propagates into the regions with high daytime NO$_2$ during the night once there is no longer any NO$_2$ assimilation to constrain the error. As a result, the night time average RMSE of CHEM is very close to that of FREE, independent of the improvement of wind simulations from daytime. In the transition from night to the daytime, the influence of assimilating NO$_2$ observations on winds begins with the first daytime cycle. This demonstrates that the
covariance of wind and NO$_2$ develops and remains during the night.

**Figure 4:** RMSE for the U winds in the urban assimilation domain. *Top:* Free (Black) vs. CHEM (Red). *Center:* T,H (Black), T,H, CHEM (Red). *Bottom:* MET (Black), Met, CHEM (Red). Note the change in scale.





Figure 2a and b show the difference in U wind between the CHEM run and the truth at 13:00 MST on July 4 before and after assimilation. The incremental change in U wind after assimilation is plotted in Figure 2c. The difference between the truth and the prior $NO_2$ column amounts viewed by the TEMPO simulator is also shown in Figure 2d. Because the U wind is underestimated in the prior, the modeled $NO_2$ plume in the prior is more concentrated at the source and more dispersed to the east than in the truth. After assimilation of the TEMPO $NO_2$ columns, we observe that the wind increases at the top and middle of the domain, where it was most underestimated prior to assimilation. Averaged over the domain, the U wind RMSE is reduced from 2.32 to 1.56 m/s from the prior to the posterior.

We assimilate observations of temperature and humidity in T,H run to adjust the wind variables. As shown in Figure 5, (T,H) shows 13.91% and 15.10% error reduction in posterior U and V winds during daytime compared to the unconstrained run FREE. These are improvements to winds from assimilating temperature and humidity observations using the covariances between meteorological variables. In addition, we find the averaged daytime posterior wind RMSE of (T,H) is very close to that of CHEM run. This demonstrates that TEMPO $NO_2$ columns, as indirect chemical observations of winds, can be used to constrain winds as well as temperature and humidity observations which are also indirect observations of winds. However, the temporal variation of the daytime posterior wind RMSE between the two runs are different (Figure 4 center). At the beginning of the daytime cycles, the T,H run shows lower posterior wind RMSE than CHEM as temperature and humidity observations are assimilated during the night, resulting in lower night time wind errors whereas no night time $NO_2$ TEMPO observations are available to be assimilated. In the later daytime cycles, the posterior wind RMSE in CHEM becomes lower than that in (T,H) due to the assimilations of TEMPO $NO_2$.

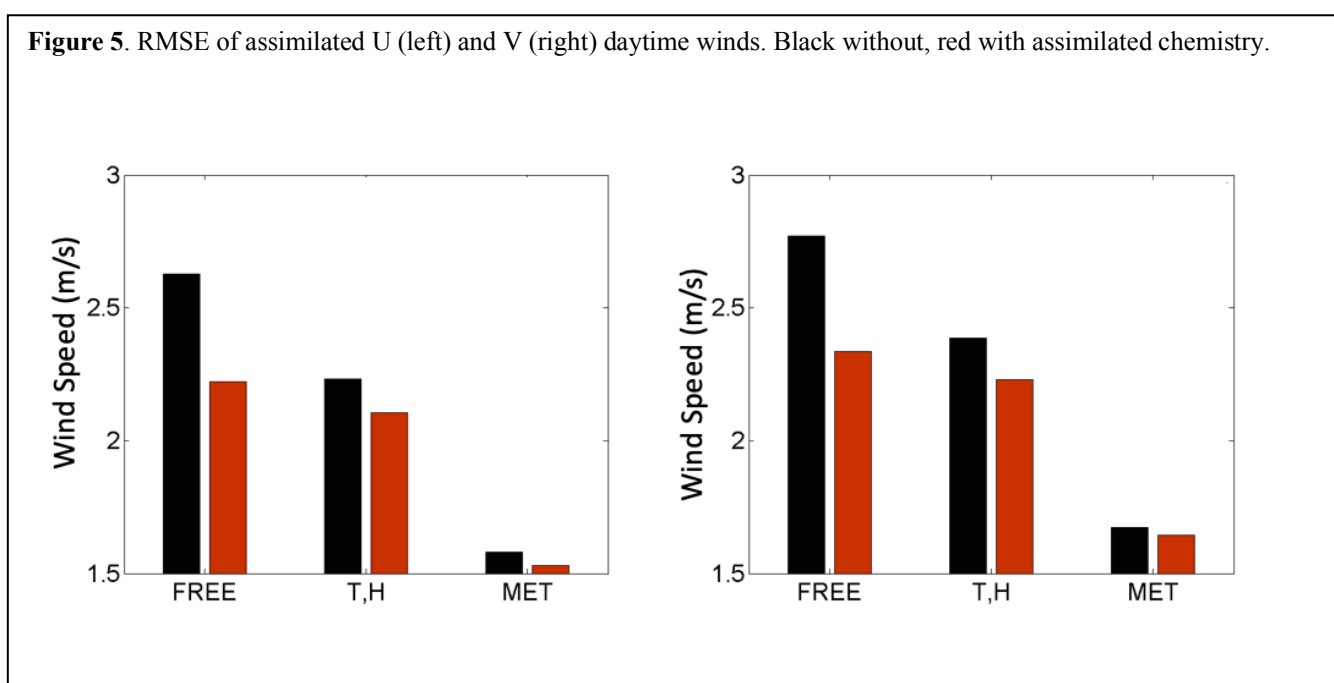

**Figure 5**. RMSE of assimilated U (left) and V (right) daytime winds. Black without, red with assimilated chemistry.

When we assimilate TEMPO $NO_2$ together with temperature and humidity observations in T,H,CHEM, we find further error reductions in posterior wind during the third day compared with T,H (Figure 4center). This is because "T,H" shows no error reductions in posterior winds in the afternoon of the third day while assimilation of TEMPO $NO_2$ alone can successfully reduce wind errors (Figure 4 top). There are only minor differences between the T,H and T,H,CHEM runs during the second daytime. This is because assimilating temperature and humidity observations alone has reduced the wind errors to the extent that assimilations of additional $NO_2$ observations can't provide further improvements. Furthermore, Figure 6 shows the wind speed in the afternoon is mostly between 2 to 4 m/s on the second day (July 4) and 4 ~ 6 m/s on the third day (July 5). When the wind is stagnant, we don't expect strong covariances between winds and $NO_2$ because the horizontal transport of $NO_2$ due to wind is not strong. When wind speed is higher on the third day, it increases the ensemble covariances between wind and $NO_2$ to achieve further improvement on wind.

The experiments MET, has the lowest RMSE in the prior estimates of $NO_2$ because it has the lowest wind errors and thus $NO_2$ transport errors as a result of the assimilation of direct wind observations (Figure 4





bottom and 5). Nevertheless, even in this run there is a small benefit to assimilating NO$_2$ columns as can be seen in the reduced RMSE of the wind on the third day.

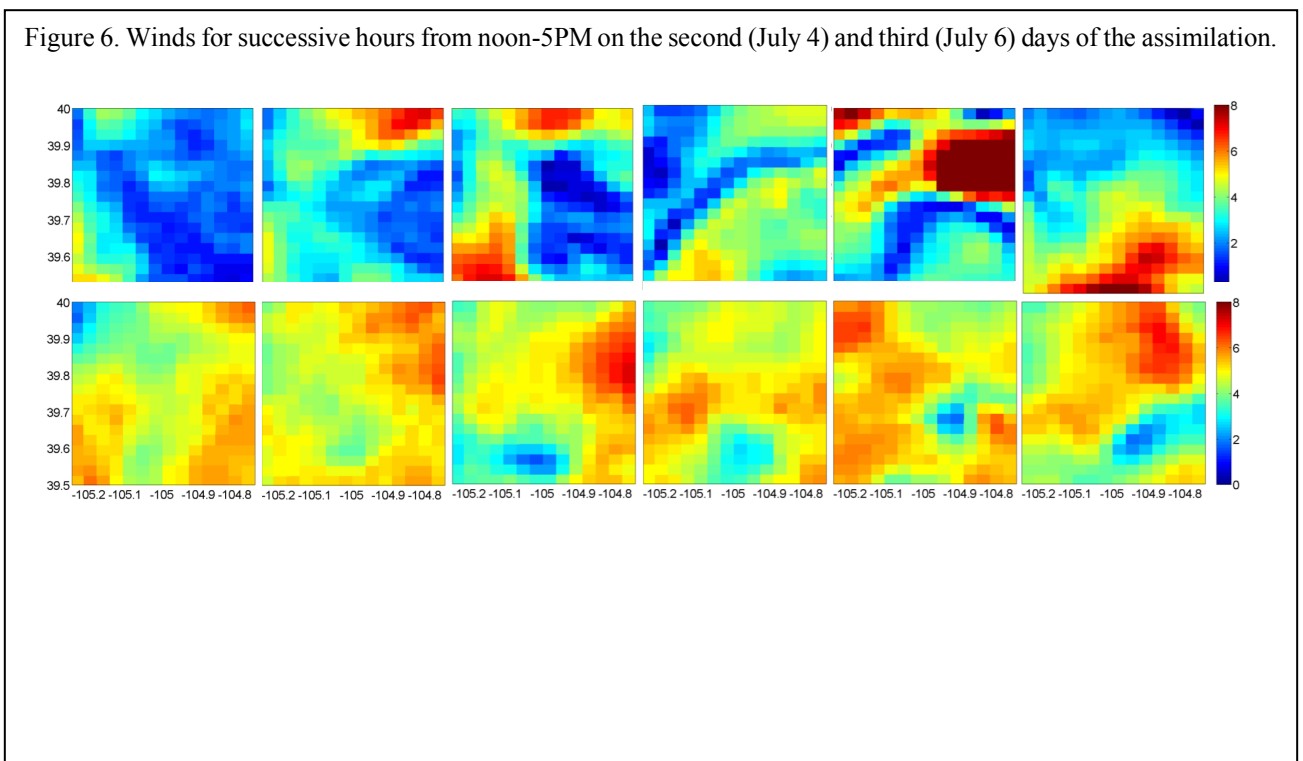

Figure 6. Winds for successive hours from noon-5PM on the second (July 4) and third (July 6) days of the assimilation.

## 5 Conclusions

Assimilation of column NO$_2$ is explored as a constraint on boundary layer winds. Compared with assimilations of temperature and humidity, assimilations of column NO$_2$ is as effective as a constraint on winds during the daytime. Column NO$_2$ which is only available in sunlight is less effective than T and H in the morning but more effective in the afternoon. In addition, we find that assimilating column NO$_2$ as will be provided by the TEMPO satellite instrument does not degrade the results of assimilating temperature and humidity observations to constrain winds, improves on wind reanalysis, especially when wind speeds are above 4 m/s. Including all available data, T, H, winds and column NO$_2$ makes it more difficult to discern the improvement from the NO$_2$ column assimilation. Nevertheless, we observe improvements in wind reanalysis even under these circumstances (Figure 5)

**Data Availability.** The data referenced in this manuscript is available to the public by visiting https://behr.cchem.berkeley.edu/home

**Author Contribution.** Xueling Liu and Ronald Cohen conceived the project, Xueling Liu developed and executed the numerical experiments. Arthur Mizzi, Jeffrey Anderson, Inez Fung and Ronald Cohen contributed to the design of the





experiments and interpretation of the results. Xueling Liu and Ronald Cohen prepared the manuscript with contributions from all co-authors.

The authors declare that they have no conflict of interest.

**Acknowledgements**. The authors gratefully acknowledge support from the NASA Grants NNX10AR36G, and NNX15AE37G and the TEMPO project grant SV3-83019. We thank N. Collins (NCAR/IMAGe) and T. Hoar (NCAR/IMAGe) for the assistance with DART. We would like to acknowledge high-performance computing support from Yellowstone (ark:/85065/d7wd3xhc) provided by NCAR's Computational and Information Systems Laboratory, sponsored by the National
Science Foundation.

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
