# Peer review of "The potential for geostationary remote sensing of NO2 to improve weather prediction"

_Atmospheric Chemistry and Physics, 2020_

## Referee Comment (RC1) · Anonymous Referee #2 · 9 Dec 2020

Review of "The potential for geostationary remote sensing of NO2 to improve weather prediction" by Xueling Liu et al. submitted to Atmospheric Chemistry and Physics.

**Summary:**

In this study, the authors use a series of short observing system simulation experiments to investigate the impact assimilating synthetic TEMPO NO2 column retrievals would have on winds in the boundary layer in a small region in western USA. The results are promising that assimilating column NO2 measurements from instruments aboard a geostationary satellite, such as the upcoming TEMPO satellite, would have a positive impact on daytime analyzed winds, though the impact is smaller (but still positive) when other meteorological observations are also included in the assimilation system.

This manuscript is suitable for publication for Atmospheric Chemistry and Dynamics; my recommendation is for this manuscript to be published after my comments and technical edits below are addressed.

**Major Comments:**

Section 2.4. Is there a reason the authors simulated their own TEMPO NO2 retrievals instead of using the synthetic TEMPO data from the TEMPO Science Team? I think it is a bit of a stretch to call this section "Observations" and encourage the authors to consider something that tells the readers that these are simulated or synthetic retrievals. Then throughout the rest of the manuscript refer to these as the "synthetic TEMPO observations" not simply "TEMPO observations". In Figure 3b & 3c captions the authors use "pseudo TEMPO NO2 observations" (but in Figure 3a they use "TEMPO column NO2 observations").

Section 4: There is room for improvement in the readability of the figures (captions and/or labels):

- For Figure 2 (Pg 8 Line 7-8) define MST relative to UTC. b has a long dash while the others say minus.
- For Figure 3, it seems that these figures could be condensed to fit on one page instead of across three pages; there does not seem to be the right balance of text in Section 4.1 for a 3-page figure. Also, is the time in the caption in MST but the figure x-axis in UTC? The blue dots are not defined (nighttime cycle). If it is not possible to combine the figures into one, Figure 3b and 3c captions are above the figures instead of below the figures, which is a more traditional location for a Table title than a figure caption.
- In Figure 4, maybe it is my pdf viewers (I tried three so I'm thinking it isn't me) but I see a white box in the upper right corner of each figure where I expect a legend should be.
- Figure 5 values are not quoted in the text, instead percent differences are referenced based on Figure 5. I recommend the authors consider either a table of the RMSE values and the percent differences or possibly a table inset of the percent differences in each panel of Figure 5.
- Figure 6 would benefit from titles along the top stating each hour and the left-hand side could label the date. If the authors do not want to include titles and row labels, then at least put panel labels (a-f, g-l) and state in the Figure caption "(July 4; a-f)" and "(July 6, g-l)" or put "(July 4; top row)" and "(July 6, bottom row)". Are the winds in Figure 6 the

U component? On Page 15, Line 11-12, the text quotes "Figure 6 shows the wind speed in the afternoon .... second day (July 4) and ... third day (July 5)". The authors need to correct the caption from July 6 to July 5.

Section 5 conclusions paint a picture that sounds very broad, whereas the OSSEs were done for a small region of the US during a few days in July 2014. This fact should be reiterated in the conclusions. Do the authors have confidence that this result would hold true under other meteorological conditions, such as winter in Salt Lake City with the strong inversions or along coastal regions where there might be land-sea breeze circulations. Are there future plans to test under different meteorological conditions and regions of the US? Are the authors working towards this method being adopted by other modeling research and operational centers?

**Minor comments/Technical edits:**

I found the chronological most recent to oldest ordering of the citations non-traditional and distracting in the Introduction. But then on Page 5 Line 6, the references are older to newer, which is my preference, and then this is the order used for the rest of the paper. The authors need to be consistent in their referencing style.

Pg 2 Line 13: "With the exception of water vapor and ozone (e.g. Inness et al., 2019), observations of atmospheric constituents are generally not used in current NWP." Inness et al. 2019 does not mention assimilation of water vapor so this reference is misleading at the start. Assimilation of water vapor is generally tropospheric only in NWP systems, only a couple groups have tested assimilation of stratospheric water vapor. Plus there are constituents assimilation systems, see Inness et al., 2013, Flemming et al., 2017 or Inness et al., 2015 (see Inness et al. 2019 references) which describes the MACC reanalysis, CAMS reanalysis and the ECMWF's Composition-IFS, respectively, which include multi-species assimilations. The authors should also specify here assimilation of trace gases or aerosols, because more NWP systems are assimilating aerosol optical depth and ozone.

Pg 2 Line 20: "....WRF-Chem have evolved rapidly in recent years (Grell and Baklanov, 2011)" Can the authors provide more references that are more recent than a single paper from nearly ten years ago.

Pg 3 Line 13: can you change this from "50~75 km" to "50 to 75 km". On the order of implies that this is an estimate.

Pg 4 Line 20: There is an extra space after "TEMPO)"

Pg 4 Lines 5-6: If I understand this correctly, the white box region in Figure 1 is the inner domain. What is the red box in Figure 1? It is not clear from the text description or the Figure 1 caption.

Pg 4 Line 7: What is the weather reanalysis data specifically? Is it the WACCM forecasts? These details are likely in the Liu et al. (2017) paper but this should be included here as well. Pg 4 Line 10: Is it worth describing why you choose NEI 2011 over 2014?

Pg 5 Line 10: "Gaspari and Cohn weighting function" does not have the year or paper reference. I presume it is 1999 from the https://doi.org/10.1002/qj.49712555417. The reference needs

fixing in the reference list (Pg 18 Line 34) as the last names are repeated "Gaspari, G., Gaspari, G., Cohn, S. E. and Cohn, S. E.:"

Pg 6 Line 3: Consider defining at least the variables W, Q, PH, MU, and PSFC. U and V are defined on Line 10.

Pg 6 Line 5: Liu et al 2017. Add period after al. and put the year in brackets.

Pg 6 Line 18: NCEP is not defined. Which NCEP dataset are the authors referring to here? Line 20 mentions GFS (not defined) forecasts.

Pg 6 Line 38: 2014060100. I would prefer to read this as "June 1, 2014 00UTC (2014060100)". Similarly on Pg 7 "2014070215" could be written out as "July 2, 2014 15UTC (2014070215)"

Pg 6 Line 38 – Pg 7 Line 2: Here, the authors reference the 12 km domain as "do1" and the 3 km domain as "d02". Can these labels be used in Figure 1?

Pg 7 Line 1: As I asked in Section 2.1, is the MOZART run something publicly available (e.g., WACCM?)

Pg 7 Line 6: NARR is not defined or described in detail. How is it different/similar to NAM that it can be used as a second meteorological driver?

Pg 7 Lines 5-6: For the control run, state the chemical boundary conditions are the same from d01. Would a pure control run use the same meteorological conditions as the NR but not allow the NO2 observations influence U and V? I agree with the authors that a second run with different reanalysis meteorology will provide an estimate of uncertainty "in the NO2 transport processes" but I do not think control run is the appropriate term.

Pg 7 Line 8: why is the assimilation starting at 13 UTC on July 3rd, not quite a day from when the NR was initialized (2014070313) and ending approximately 2.5 days later (2014070600). Pg 7 Line 9: the authors do not need to define NR again.

Pg 7 Line 22: add synthetic to "we assimilate synthetic TEMPO NO2 observations"

Pg 7 Line 24: Change to "In the next two experiments (hereafter, T and H, respectively)". I found H confusing here as I am used to H for geopotential height and either Q or RH for humidity. I suggest the authors reconsider changing the H to Q.

Pg 7 Lines 26, 32: to be consistent with Figure 3, change to T,H + CHEM and MET+CHEM. Also change this in Figure 4 caption.

Pg 7 Line 36, 38: change the "/n" to  $n^{-1}$  and "/l" to "l-1"

Pg 7 Line 38, Pg 8 Line 2: be consistent throughout manuscript as either Figure or Fig.

Pg 10 Line 7: If you exclude the first day from your analysis, it could help the reader if you greyed out in Figures 3 and 4 the time period you consider still spin up for assimilation system. Pg 10 Line 9: The statement "CHEM shows error reduction in the posterior wind analysis in each daytime cycle" should be labeled as Figure 4 top panel in the main text. The discussion of the center and bottom panels of Figure 4 is disconnected from this paragraph by a discussion of Figure 2 again so be sure to state Figure 4 "top panel" in this paragraph.

Pg 14 Line 8: These numbers are not in Figure 5. Are the values in Figure 5 for the same domain?

Pg 14 Line 14: It is not clear to me if "Chem run" is in reference to the red bar above the "FREE" or "T,H". At first, I read it as it must be the CHEM from the FREE but then the authors discuss Figure 4 center panel. Then line 4 of Page 15 the authors discuss T,H,CHEM so I am struggling with the naming conventions of the experiment runs and the reference to them within the text.

Pg 16 Line 14: The Conclusion ends without a period.

Pg 19 Line 19: Fix doi reference to have the h for https.

Given I found two errors in the reference list on the two I checked (Gaspari and Cohn, and Inness et al.), the authors should go over each entry and make sure there are no further errors.

---

## Referee Comment (RC2) · Anonymous Referee #3 · 23 Dec 2020

Title: The potential for geostationary remote sensing of NO2 to improve weather prediction Xueing Liu et al., 2020

Review: Summary: This manuscript examines the impact of the assimilation of hourly observations of tropospheric NO2 columns on near surface winds. Using a series of assimilation experiments over the Denver, CO region, and synthetic TEMPO observations, they quantify the impact of the NO2 assimilation with respect to the assimilation of different meteorological parameters. They find that while the assimilation of NO2 improves the representation of boundary layer winds, that improvement is of the same magnitude as when the meteorology alone is assimilated. When the meteorology and NO2 are both assimilated, the improvement from including the NO2 observations is marginal.

I recommend this manuscript for publication with minor revisions as described below.

Major comments: The series of experiments described are over very small temporal and spatial scales. The applicability of these results to other regions and seasons is not discussed. Understanding that repeating the analysis for multiple regions and seasons is beyond the scope of the manuscript, the conclusions should be refined to acknowledge this.

Make sure references are cited correctly and in a consistent style throughout the manuscript.

Throughout the manuscript the authors refer to 'TEMPO NO2'. This should be changed to more accurately indicate that synthetic data is being used.

Minor comments:

P2L12: "With the exception of water vapor and ozone (e.g. Inness et al., 2019), observations of atmospheric constituents are generally not used in current NWP." There are a host of operational global models that assimilate observations of aerosol optical depth. See: Xian, et al. Current state of the global operational aerosol multi-model ensemble: An update from the International Cooperative for Aerosol Prediction (ICAP). Q J R Meteorol Soc 2019 doi:10.1002/qj.3497

P3L3: "Examples in simpler models include studies by Allen et al., 2014, 2015; and Haussaire and Bocquet, 2015." This sentence likely needs to be fleshed out.

P4L7: Specify reanalysis data used for initial and boundary conditions

Figure 1: The figure caption doesn't mention what the red square inside the inner domain represents. State boundaries would also help, given that the inner domain location is referenced with respect to Denver.

P6L4: Fix Romine reference

Section 2.3: I suggest explaining this more accessibly and less in terms of namelist
options and code. E.g. explain what T170 resolution is, and any specific code or namelist options can be put in a table in supplemental material.

Section 2.4: Consider changing the title of this section to 'Synthetic Observations' or something like that. What is the reasoning for using the author's own TEMPO simulator, rather than the synthetic data provided by the TEMPO science team?

P6-P7: "The meteorological initial and boundary conditions are taken from the North American Mesoscale Forecast System (NAM), and the chemistry simulation is constrained by MOZART output." What does this mean? That MOZART was used as initial/boundary conditions for this simulation?

P7L4: Spell out NARR, and potentially mention the differences between the two reanalyses (e.g. at least that NAM has output at 12km spatial resolution, and NARR has 32km).

P8L1/Figure2: Is this an average of the lowest 5 levels? It would be useful to know what average altitude range this translates too. E.g. is it possible that power plant towers, which are important emission sources for NO2, could be above this level? Also, the colorbar legend for Figure 2d is too small to read.

Figure 3: Y axes are too small to read, and X axes should specify time zone. These sets of figures could probably go on one page, rather than three.

Figure 4: There is a white box in the upper right corner of each panel. Given that these panels are described with respect to each other, consider putting them all on the same scale to allow for easier comparison.

P15L16: Should be 'The experiment MET has the..." I think this sentence might missing a word. It reads like a fragment.

Figure 6: Bigger labels, label the hours, and larger color bar labels with units.

2020.

---

## Author Comment (AC1) · 27 Feb 2021

We thank both reviewers for their positive and constructive comments.

All of the editorial/typographical comments will be addressed in the revision.

Reviewer comments are italicized and our response indented.

**Reviewer #3**

Major comments: The series of experiments described are over very small temporal and spatial scales. The applicability of these results to other regions and seasons is not discussed. Understanding that repeating the analysis for multiple regions and seasons is beyond the scope of the manuscript, the conclusions should be refined to acknowledge this. Make sure references are cited correctly and in a consistent style throughout the manuscript.

We have added comments about the need for additional research to assess the utility of this assimilation approach in other cities and seasons.

Throughout the manuscript the authors refer to 'TEMPO NO2'. This should be changed to more accurately indicate that synthetic data is being used.

Done

Minor comments:

P2L12: "With the exception of water vapor and ozone (e.g. Inness et al., 2019), observations of atmospheric constituents are generally not used in current NWP." There are a host of operational global models that assimilate observations of aerosol optical depth. See: Xian, et al. Current state of the global operational aerosol multimodel ensemble: An update from the International Cooperative for Aerosol Prediction (ICAP). Q J R Meteorol Soc 2019 doi:10.1002/qj.3497

Noted and reference added

*P3L3: "Examples in simpler models include studies by Allen et al., 2014, 2015; and Haussaire and Bocquet, 2015." This sentence likely needs to be fleshed out.*

Additional references added

P4L7: Specify reanalysis data used for initial and boundary conditions Figure 1: The figure caption doesn't mention what the red square inside the inner domain represents. State boundaries would also help, given that the inner domain location is referenced with respect to Denver.

Text added.

Section 2.3: I suggest explaining this more accessibly and less in terms of namelist options and code. E.g. explain what T170 resolution is, and any specific code or namelist options can be put in a table in supplemental material.

Done.

Section 2.4: Consider changing the title of this section to 'Synthetic Observations' or something like that. What is the reasoning for using the author's own TEMPO simulator, rather than the synthetic data provided by the TEMPO science team?

Title changed as suggested.

We developed our simulator prior to the availability of the TEMPO science team's work. Also, we anticipated it would be advantageous to have full control over the parameters chosen in simulator in-house.

P6-P7: "The meteorological initial and boundary conditions are taken from the North American Mesoscale Forecast System (NAM), and the chemistry simulation is constrained by MOZART output." What does this mean? That MOZART was used as initial/boundary conditions for this simulation?

We have tried to clarify. NAM was used to initialize the meteorology and MOZART the concentrations of chemicals.

*P8L1/Figure2: Is this an average of the lowest 5 levels? It would be useful to know what average altitude range this translates too. e.g. is it possible that power plant towers, which are important emission sources for NO2, could be above this level? Also, the color bar legend for Figure 2d is too small to read.*

Text added to provide a fuller description. And color bar size increased.

*Figure 3: Y axes are too small to read, and X axes should specify time zone. These sets of figures could probably go on one page, rather than three.*

We will improve readability and work with ACP production to try to get them on 1 page.

Figure 4: There is a white box in the upper right corner of each panel. Given that these panels are described with respect to each other, consider putting them all on the same scale to allow for easier comparison.

The production version of these figures did not view exactly as intended. We will revise to improve the rendering. Instead of changing the scale we have called out the different scales more explicitly in the text.

*P15L16:* Should be 'The experiment MET has the: : : " I think this sentence might missing a word. It reads like a fragment.

Fixed

Figure 6: Bigger labels, label the hours, and larger color bar labels with units.

Addressed as suggested

**Review #2**

Major Comments:

Section 2.4. Is there a reason the authors simulated their own TEMPO NO2 retrievals instead of using the synthetic TEMPO data from the TEMPO Science Team?

As noted above in response to the other reviewer, we developed our simulator prior to the availability of the TEMPO science team's work. Also, we anticipated it would be advantageous to have full control over the parameters chosen in simulator in-house.

I think it is a bit of a stretch to call this section "Observations" and encourage the authors to consider something that tells the readers that these are simulated or synthetic retrievals. Then throughout the rest of the manuscript refer to these as the "synthetic TEMPO observations" not simply "TEMPO observations". In Figure 3b & 3c captions the authors use "pseudo TEMPO NO2 observations" (but in Figure 3a they use "TEMPO column NO2 observations").

Similar comments were made by the other reviewer. We have changed the title of this section and clarified the reference to simulated data throughout as suggested by the reviewer.

Section 4: There is room for improvement in the readability of the figures (captions and/or labels):

• For Figure 2 (Pg 8 Line 7-8) define MST relative to UTC. b has a long dash while the others say minus. Done

• For Figure 3, it seems that these figures could be condensed to fit on one page instead of across three pages; there does not seem to be the right balance of text in Section 4.1 for a 3-page figure.

We will work with production to fit the figures into a single page

*Also, is the time in the caption in MST but the figure x-axis in UTC? The blue dots are not defined (nighttime cycle).*

Fixed.

If it is not possible to combine the figures into one, Figure 3b and 3c captions are above the figures instead of below the figures, which is a more traditional location for a Table title than a figure caption.

Fixed

• In Figure 4, maybe it is my pdf viewers (I tried three so I'm thinking it isn't me) but I see a white box in the upper right corner of each figure where I expect a legend should be.

Fixed.

• Figure 5 values are not quoted in the text, instead percent differences are referenced based on Figure 5. I recommend the authors consider either a table of the RMSE values and the percent differences or possibly a table inset of the percent differences in each panel of Figure 5.

We have replaced Figure 5 with a Table.

• Figure 6 would benefit from titles along the top stating each hour and the left-hand side could label the date. If the authors do not want to include titles and row labels, then at least put panel labels (a-f, g-l) and state in the Figure caption "(July 4; a-f)" and "(July 6, g-l)" or put "(July 4; top row)" and "(July 6, bottom row)". Are the winds in Figure 6 the U component? On Page 15, Line 11-12, the text quotes "Figure 6 shows the wind speed in the afternoon .... second day (July 4) and ... third day (July 5)". The authors need to correct the caption from July 6 to July 5.

**Changes made as suggested by the referee.**

Section 5 conclusions paint a picture that sounds very broad, whereas the OSSEs were done for a small region of the US during a few days in July 2014. This fact should be reiterated in the conclusions. Do the authors have confidence that this result would hold true under other meteorological conditions, such as winter in Salt Lake City with the strong inversions or along coastal regions where there might be land-sea breeze circulations.

We have added text to the conclusions that highlight that this study points to the promise of this approach and further evaluation under other meteorological conditions such as those suggested by the referee should be developed and evaluated.

Are there future plans to test under different meteorological conditions and regions of the US? Are the authors working towards this method being adopted by other modeling research and operational centers?

At this time we have no specific plans to test the method. We hope this paper will inspire operational centers to consider the benefits of chemical data assimilation.

**Minor comments/Technical edits:**

I found the chronological most recent to oldest ordering of the citations non-traditional and distracting in the Introduction. But then on Page 5 Line 6, the references are older to newer, which is my preference, and then this is the order used for the rest of the paper. The authors need to be consistent in their referencing style.

Done.

*Pg 2 Line 13: "With the exception of water vapor and ozone (e.g. Inness et al., 2019), observations of atmospheric constituents are generally not used in current NWP." Inness et al. 2019 does not mention assimilation of water vapor so this reference is misleading at the start.*

Sentence is fixed to clarify.

Assimilation of water vapor is generally tropospheric only in NWP systems, only a couple groups have tested assimilation of stratospheric water vapor. Plus there are constituents assimilation systems, see Inness et al., 2013, Flemming et al., 2017 or Inness et al., 2015 (see Inness et al. 2019 references) which describes the MACC reanalysis, CAMS reanalysis and the ECMWF's Composition-IFS, respectively, which include multi-species assimilations. The authors should also specify here assimilation of trace gases or aerosols, because more NWP systems are assimilating aerosol optical depth and ozone.

Clarified in the text.

*Pg 2 Line 20: "….WRF-Chem have evolved rapidly in recent years (Grell and Baklanov, 2011)" Can the authors provide more references that are more recent than a single paper from nearly ten years ago.*

We removed the statement about rapid evolution.

Pg 3 Line 13: can you change this from "50~75 km" to "50 to 75 km". On the order of implies that this is an estimate.

Done

Pg 4 Line 20: There is an extra space after "TEMPO)"

Done

*Pg 4 Lines 5-6: If I understand this correctly, the white box region in Figure 1 is the inner domain. What is the red box in Figure 1? It is not clear from the text description or the Figure 1 caption.*

Text added to the caption to clarify.

*Pg 4 Line 7: What is the weather reanalysis data specifically? Is it the WACCM forecasts? These details are likely in the Liu et al. (2017) paper but this should be included here as well.*

Reference to NAM meteorology used is on page 6 and will be moved here as well

*Pg 4 Line 10: Is it worth describing why you choose NEI 2011 over 2014?*

We don't think this point helps in evaluating the method and refrain from adding text.

*Pg* 5 *Line* 10: "Gaspari and Cohn weighting function" does not have the year or paper reference. I presume it is 1999 from the https://doi.org/10.1002/qj.49712555417. The reference needs fixing in the reference list (*Pg* 18 *Line* 34) as the last names are repeated "Gaspari, G., Gaspari, G., Cohn, S. E. and Cohn, S. E.:"

Done

*Pg* 6 *Line* 3: *Consider defining at least the variables W, Q, PH, MU, and PSFC. U and V are defined on Line 10.*

Done

Pg 6 Line 5: Liu et al 2017. Add period after al. and put the year in brackets.

Done

*Pg* 6 *Line* 18: *NCEP is not defined. Which NCEP dataset are the authors referring to here? Line* 20 *mentions GFS (not defined) forecasts.*

Defined NCEP and GFS acronyms.

Pg 6 Line 38: 2014060100. I would prefer to read this as "June 1, 2014 00UTC (2014060100)".

Done

Similarly on Pg 7 "2014070215" could be written out as "July 2, 2014 15UTC (2014070215)"

Done

Pg 6 Line 38 - Pg 7 Line 2: Here, the authors reference the 12 km domain as "do1" and the 3 km domain as "d02". Can these labels be used in Figure 1?

Done

Pg 7 Line 1: As I asked in Section 2.1, is the MOZART run something publicly available (e.g., WACCM?)

Yes the run is publicly available, but no MOZART and WACCM are different models. We added a pointer to the MOZART archive.

*Pg* 7 *Line* 6: *NARR* is not defined or described in detail. How is it different/similar to NAM that it can be used as a second meteorological driver?

The NARR and NAM are two independent meteorological analyses.

*Pg* 7 Lines 5-6: For the control run, state the chemical boundary conditions are the same from d01. Would a pure control run use the same meteorological conditions as the NR but not allow the NO2 observations influence U and V? I agree with the authors that a second run with different reanalysis meteorology will provide an estimate of uncertainty "in the NO2 transport processes" but I do not think control run is the appropriate term.

We prefer to keep this labelling as it does convey the intent to use this as a point of comparison for the other calculations.

*Pg* 7 *Line* 8: *why is the assimilation starting at* 13 *UTC on July* 3*rd, not quite a day from when the NR was initialized (2014070313) and ending approximately* 2.5 *days later (2014070600).*

This timing allows analyses of 3 daytime periods. Text added to explain this point.

Pg 7 Line 9: the authors do not need to define NR again.

Done

Pg 7 Line 22: add synthetic to "we assimilate synthetic TEMPO NO2 observations"

Done

*Pg* 7 Line 24: Change to "In the next two experiments (hereafter, T and H, respectively)". I found H confusing here as I am used to H for geopotential height and either Q or RH for humidity. I suggest the authors reconsider changing the H to Q.

We changed H to RH.

*Pg* 7 *Lines* 26, 32: to be consistent with Figure 3, change to T,H + CHEM and MET+CHEM. Also change this in Figure 4 caption.

Done

*Pg* 7 *Line* 36, 38: *change the "/n" to n-1 and "/l" to "l-1"*

Done

Pg 7 Line 38, Pg 8 Line 2: be consistent throughout manuscript as either Figure or Fig.

Done

*Pg* 10 *Line* 7: *If you exclude the first day from your analysis, it could help the reader if you greyed out in Figures 3 and 4 the time period you consider still spin up for assimilation system.*

All of the data shown is included in our analysis

Pg 10 Line 9: The statement "CHEM shows error reduction in the posterior wind analysis in each daytime cycle" should be labeled as Figure 4 top panel in the main text. The discussion of the center and bottom panels of Figure 4 is disconnected from this paragraph by a discussion of Figure 2 again so be sure to state Figure 4 "top panel" in this paragraph.

Additional pointers to the figures we are discussing are added to the text as suggested.

Pg 14 Line 8: These numbers are not in Figure 5. Are the values in Figure 5 for the same domain?

The values for figure 5 are for the same domain.

Pg 14 Line 14: It is not clear to me if "Chem run" is in reference to the red bar above the "FREE" or "T,H". At first, I read it as it must be the CHEM from the FREE but then the authors discuss Figure 4 center panel. Then line 4 of Page 15 the authors discuss T,H,CHEM so I am struggling with the naming conventions of the experiment runs and the reference to them within the text.

We have added text to clarify.

*Pg* 16 *Line* 14: *The Conclusion ends without a period.*

Done

*Pg* 19 *Line* 19: *Fix doi reference to have the h for* **h***ttps.*

Done

Given I found two errors in the reference list on the two I checked (Gaspari and Cohn, and Inness et al.), the authors should go over each entry and make sure there are no further errors

All the references have been checked.